# Characterization of *Pseudomonas* sp. En3, an Endophytic Bacterium from Poplar Leaf Endosphere with Plant Growth-Promoting Properties

Beiyan Deng, Ling Wu, Hongju Xiao and Qiang Cheng *

State Key Laboratory of Tree Genetics and Breeding, Key Laboratory of Forestry Genetics & Biotechnology of Ministry of Education, Co-Innovation Center for Sustainable Forestry in Southern China,
Nanjing Forestry University, Nanjing 210037, China; dengbeiyan@njfu.edu.cn (B.D.); wuling@njfu.edu.cn (L.W.); xiaohongju@njfu.edu.cn (H.X.)
* Correspondence: chengqiang@njfu.edu.cn

**Abstract:** Growth-promoting endophytic bacteria possess substantial potential for sustainable agriculture. Here, we isolated an endophytic bacterium, *Pseudomonas* sp. En3, from the leaf endosphere of *Populus tomentosa* and demonstrated its significant growth-promoting effects on both poplar and tomato seedlings. The phosphorus solubilization and nitrogen fixation abilities of strain En3 were confirmed via growth experiments on NBRIP and Ashby media, respectively. Salkowski staining and HPLC-MS/MS confirmed that En3 generated indole-3-acetic acid (IAA). The infiltration of En3 into leaf tissues of multiple plants did not induce discernible disease symptoms, and a successful replication of En3 was observed in both poplar and tobacco leaves. Combining Illumina and Nanopore sequencing data, we elucidated that En3 possesses a circular chromosome of 5.35 Mb, exhibiting an average G + C content of 60.45%. The multi-locus sequence analysis (MLSA) and genome average nucleotide identity (ANI) supported that En3 is a novel species of *Pseudomonas* and constitutes a distinct phylogenetic branch with *P. rhizosphaerae* and *P. coleopterorum*. En3 genome annotation analysis revealed the presence of genes associated with nitrogen fixation, phosphate solubilization, sulfur metabolism, siderophore biosynthesis, synthesis of IAA, and ethylene and salicylic acid modulation. The findings suggest that *Pseudomonas* sp. En3 exhibits significant potential as a biofertilizer for crop and tree cultivation.

**Keywords:** poplar; endophytes; plant growth promotion; *Pseudomonas*; colonization; genome





## 1. Introduction

Endophytic bacteria can colonize plant tissues without eliciting any disease symptoms in host plants [1]. Many endophytic bacteria can promote plant growth by assisting the acquisition of essential nutrients, synthesizing phytohormones, or enhancing host resistance against pathogenic microorganisms. The mechanism of plant growth promotion via endophytic bacteria resembles that of rhizosphere bacteria residing in soil; however, their presence within plant tissues allows for more direct beneficial effects on the closely associated plant cells [2]. Therefore, the identification and characterization of endophytic plant growth-promoting (PGP) bacteria holds significant potential for application in sustainable agriculture and silviculture [3].

*Pseudomonas* is a diverse and complex bacterial genus that occurs in a broad range of ecological niches, including plant tissues [4]. Some endophytic *Pseudomonas* species are well known as PGP agents. For instance, the rice endophytic *P. stutzeri* A15 employs a nitrogenase complex to fix nitrogen [5], and three *Pseudomonas* isolates derived from *Miscanthus giganteus* can produce gluconic acid to solubilize inorganic phosphate compounds [6]. *P. mosselii* and *P. putida* W619 from the root tissues of wheat and poplar were confirmed to biosynthesize indole-3-acetic acid (IAA) via the tryptophan pathway [7,8].

*P. fluorescens* YsS6 and *P. migulae* 8R6 can reduce ethylene content in mini carnation fresh cut flowers via 1-aminocyclopropane-1-carboxylate (ACC) deaminase [9]. *P. fluorescens* BRZ63, *P. simiae* PICF7, and *P. putida* BP25, endophytes of oilseed rape, olive, and pepper, respectively, have been found to inhibit the growth of various pathogenic fungi or bacteria [10–12]. The endophytic bacterium *P. brassicacearum* Zy-2-1 from *Sphaerophysa salsula*, *P. fluorescens* Sasm05 from *Sedum alfredii* Hance, and *P. azotoformans* ASS1 from *Alyssum serpyllifolium* can protect plants from abiotic stress and expedite the phytoremediation process of metal-contaminated soils [13–15]. The colonization of host tissues by these endophytic *Pseudomonas* species continuously confers benefits to their hosts and presents significant potential for application as alternatives to chemical fertilizers and pesticides.

Poplar trees (*Populus* spp.) are widely cultivated worldwide for producing fiberboard, paper pulp, and construction lumber [16]. Several poplar endophytic bacteria derived from root and stem endosphere have been characterized for their growth-promoting abilities. Two strains from different *Burkholderia* species, *B. vietnamiensis* WPB and *B. pyrrocinia* JK-SH007, both isolated from poplar stem, can promote growth of bluegrass and poplar seedlings, respectively [17–19]. *Enterobacter* sp. strain 638 and *P. putida* W619 are primarily colonized in poplar roots and have been shown to promote biomass accumulation of poplar cuttings [8,20]. To date, there are no reports characterizing PGP bacteria from the poplar leaf endosphere.

In this study, *Pseudomonas* sp. En3 was isolated from the leaf endosphere of poplar, which can grow in the leaves of multiple plants without causing disease symptoms and promote the growth of poplar and tomato plants. Strain En3 exhibits auxin production, phosphate solubilization, and nitrogen fixation capabilities. Multi-locus sequence analysis (MLSA) and genome-wide comparison revealed it as a novel *Pseudomonas* species. Genome annotation analysis indicates its high potential for promoting plant growth.

## 2. Materials and Methods

### 2.1. Isolation of Endophytic Bacteria from Poplar Leaves

In July 2022, healthy leaves of *Populus tomentosa* within the campus of Nanjing Forestry University in Nanjing, Jiangsu Province, China (118.82° E, 32.08° N) were collected. The leaf discs were immersed in a 5% NaClO solution for 5 min and subsequently rinsed five times with sterile distilled water. The surface-disinfected leaf discs were placed on LB agar plates at 28 °C for 48 h. Bacteria observed at the periphery of the leaf discs were streaked, and single colonies were preserved. To validate the efficacy of the sterilization procedure, the final rinse water (100 μL) was spread onto LB agar plates and incubated under the same conditions.

### 2.2. Identification of Endophytic Bacteria

Genomic DNA of endophytic bacteria was extracted using the TIANamp Bacteria DNA Kit (Tiangen, Beijing, China). The 16S rRNA gene was amplified using primer pairs 27F (5′-AGAGTTTGATCMTGGCTCAG-3′) and 1492R (5′-GGTTACCTTGTTACGACTT-3′) [21]. PCR products were purified from the gel and ligated into the pMD19-T vector (Takara, Dalian, China) for Sanger sequencing. The obtained 16S rDNA sequences were subjected to a BLASTn search against the NCBI rRNA database and ezBiocloud (https://www.ezbiocloud.net/, accessed on 26 October 2023).

### 2.3. Plant Growth-Promoting Assays

The poplar cultivar 'Shanxin' (*Populus davidiana* × *P. bolleana*) was micropropagated on half-strength Murashige and Skoog (MS) medium. Two-week-old rooted seedlings were transplanted into pots containing nutrient soil and subsequently cultivated in a growth chamber under controlled conditions of a 16 h light/8 h dark photoperiod at 23 °C. One week post-transplantation, the seedlings were used in plant growth-promoting assays. Tomato plants (*Solanum lycopersicum*) were sowed under identical conditions in the same growth chamber; one-week-old tomato seedlings were used for plant growth-promoting

assays. Bacterial cells were collected from LB plates and resuspended in sterile distilled water (OD600 = 0.5). The bacterial suspension was administered twice weekly (30 mL per pot), while control plants received an equal volume of sterile distilled water. The plant height (from apical meristem to the soil substrate) and fresh weight (whole plant after washing off soil substrate and blotting out water) of the plants were measured and recorded.

### 2.4. Plant Growth-Promoting Properties of Strain En3

The phosphate solubilizing activities of strain En3 were assessed by spotting 10 μL of cultures on the top of NBRIP plates (10 g D-glucose, 5 g $Ca_3(PO_4)_2$, 5 g $MgCl_2 \cdot 6H_2O$, 0.25 g $MgSO_4 \cdot 7H_2O$, 0.2 g KCl, 0.1 g $(NH_4)_2SO_4$, 15 g agar in 1 L distilled water) [22]. The presence or absence of a transparent halo around bacterial colonies was recorded.

To assess nitrogen-fixing ability, a 10 μL aliquot of the culture of En3 was inoculated onto Ashby nitrogen-free agar medium (5 g glucose, 5 g mannitol, 0.1 g $CaCl_2 \cdot 2H_2O$, 0.1 g $MgSO_4 \cdot 7H_2O$, 5 mg $Na_2MoO_4 \cdot 2H_2O$, 0.9 g$K_2HPO_4$, 0.1 g $KH_2PO_4$, 0.01 g $FeSO_4 \cdot 7H_2O$, 5 g $CaCO_3$, 15 g agar in 1 L distilled water) [23].

The Salkowski method was employed to evaluate the capacity of En3 for indole-3-acetic acid (IAA) production. Briefly, the bacteria were cultured in LB liquid medium supplemented with 0.5 mg/mL L-tryptophan until reaching OD600 of 1.5. The supernatants were then mixed with Salkowski reagent (1.2% $FeCl_3$ in 37% $H_2SO_4$) for 30 min, and the resulting color reaction was observed [24]. Positive controls included a solution containing 50 mg/L IAA and *P. syringae* pv. *tomato* DC3000 supernatant. Furthermore, culture supernatants from three independent experiments were subjected to high-performance liquid chromatography and tandem mass spectrometry (HPLC-MS/MS) analysis at RaySource Biotechnology (Nanjing, China) to determine the concentrations of IAA and IAA synthesis intermediates including indole-3-acetamide (IAM), indole-3-pyruvate (IPyA), and indole-3-acetonitrile (IAN).

### 2.5. Analysis of the Colonization of Strain En3 in Multiple Plants

Strain En3 colonization experiments were conducted using three-week-old tobacco (*Nicotiana benthamiana*), two-week-old tomato, and two-week-old oilseed rape (*Brassica napus*) plants grown in nutrient soil in growth chamber. Additionally, four-week-old poplar 'Shanxin' seedlings grown on half-strength MS medium were also used. Bacterial cells were suspended in 10 mM $MgCl_2$ and diluted to a concentration of $10^5$ prior to infiltration into the leaves of tobacco, tomato, and oilseed rape using a syringe. For poplar 'Shanxin' seedlings, the bacterial suspension with 0.015% Silwet L-77 was infiltrated under continuous vacuum for 3 min at a pressure of 500 mmHg. Afterward, the seedlings were dried with sterile filter paper and planted in half-strength MS medium supplemented with 100 mg/L Cefotaxime.

One spontaneous rifampicin-resistant mutant from strain En3 was used for a growth curve assay to avoid contamination by environmental bacteria. Leaf discs were collected at 0, 2, or 4 days post-infiltration (dpi), and homogenized in 10 mM $MgCl_2$. The cfu/$cm^2$ for tobacco and cfu/mg for poplar were determined by plating serial dilutions of leaf extracts on LB plates containing 100 mg/L rifampicin. Three independent experiments were performed [25].

### 2.6. Genome Sequencing, Assembling, and Annotation

The genomic DNA of strain En3 was extracted using the TIANamp Bacteria DNA Kit (Tiangen, Beijing, China). The quality and concentration of DNA were verified on 1.5% agarose gels and using a Qubit 3.0 Fluorometer (Invitrogen, Carlsbad, CA, USA). Long-read and short-read sequencing DNA libraries were prepared with an SQK-LSK109 ligation kit (Oxford Nanopore Technologies (ONT), Oxford, UK) and Nextera DNA Flex Library Prep Kit (Illumina, San Diego, CA, USA), respectively, according to the manufacturer's protocols. The DNA libraries were then sequenced by the Oxford Nanopore PromethION sequencer (ONT) with 48 h runs and the Illumina NovaSeq 6000 platform (Illumina) with paired-end 150 bp read lengths by Benagen company (Wuhan, China). The En3 genome

was assembled using Unicycler software (Version: 0.5.0) [26] with a combination of ONT and Illumina reads. Then, two rounds of error correction were performed on the assembly result based on the Illumina reads using Pilon (v.1.23) [27].

The coding sequences, tRNAs, and rRNAs were predicted by Prodigal, Aragorn, and RNAmmer in Prokka software (Version: 1.14.6) [28]. The predicted protein functions were further annotated using the National Center for Biotechnology Information (NCBI) non-redundant proteins (NR) and Kyoto Encyclopedia of Genes and Genomes (KEGG) databases.

### 2.7. Phylogenetic and Average Nucleotide Identity (ANI) Analyses

The DNA sequences of 16S rRNA, *gyrB*, *rpoD*, and *rpoB* genes from the representative type strains of *P. fluoresens* group, *P. syringae* group, *P. angulliseptica* group, *P. alcaligenes* group, *P. oleovorans* group, and *P. putida* group [29], as well as strain En3, were concatenated and aligned. A phylogenetic tree of the multilocus alignment was constructed with the neighbor-joining method using MEGA 7.0 [30]. The inferred phylogenies were tested by 1000 bootstrap replicates.

The average nucleotide identity (ANI) was calculated using the OrthoANIu algorithm [31]. The automated multi-locus species tree (autoMLST) [32] was also used for analyzing phylogeny and whole genome similarity.

## 3. Results

### 3.1. Isolation of Endophytic Bacteria from Poplar Leaves and Analysis of Plant Growth-Promoting Activity

We isolated four strains of endophytic bacteria (En1–En4) with different colony morphology from healthy *P. tomentosa* leaves through rigorous surface disinfection procedures. The 16S rRNA genes of these strains were sequenced and compared against the NCBI rRNA database and ezBiocloud, revealing top hits *Sphingomonas sanguinis* strain NBRC 13937[T] (99.86% identity), *Sphingobium yanoikuyae* strain NBRC 15102[T] (99.72% identity), *Pseudomonas azerbaijanorientalis* strain SWRI123[T] (99.10% identity), and *Microbacterium testaceum* strain NBRC 12675[T] (99.10% identity). Consequently, we designated them as *Sphingomonas* sp. En1 (GenBank accession no. OR511661), *Sphingobium* sp. En2 (OR511660), *Pseudomonas* sp. En3 (OR511655), and *Microbacterium* sp. En4 (OR511666) (Table 1).

**Table 1.** Sequence analysis of 16S rRNA of endophytic bacteria isolated from Poplar leaves.

| Endophyte | Top Hit (Accession No.) | Sequence Length (bp) | % Identity | Genbank Accession No. |
|---|---|---|---|---|
| En1 | *Sphingomonas sanguinis* NBRC 13937[T] (NR_113637) | 1409 | 99.86 | OR511661 |
| En2 | *Sphingobium yanoikuyae* NBRC 15102[T] (NR_113730) | 1409 | 99.72 | OR511660 |
| En3 | *Pseudomonas azerbaijanorientalis* SWRI123[T] (CP077078) | 1448 | 99.10 | OR511655 |
| En4 | *Microbacterium testaceum* NBRC 12675[T] (NZ_BJMLO1000022) | 1444 | 99.10 | OR511666 |

To assess the plant growth-promoting activity of endophytic bacteria, we continuously irrigated transplanted poplar 'Shanxin' seedlings with a suspension of these four strains. After one month of treatment, none of the four strains induced any disease symptoms in poplar seedlings; only *Pseudomonas* sp. En3 exhibited a significant ability to enhance the growth of poplar trees ($p < 0.01$) (Figure 1A). Compared to the control group (irrigated with $H_2O$), En3-irrigated poplar trees displayed a substantial increase in plant height by 54% and fresh weight by 87% on average (Figure 1B,C). Furthermore, tomato seedlings treated with En3 for two weeks demonstrated remarkable growth-promoting effects ($p < 0.01$) in

both plant height (an average increase of 67%) and fresh weight (an average increase of 171.1%) compared to the control group (Figure 1D–F). The fact that *Pseudomonas* sp. En3 promotes growth in both distantly related plant species suggests the possibility of general plant-promoting features.

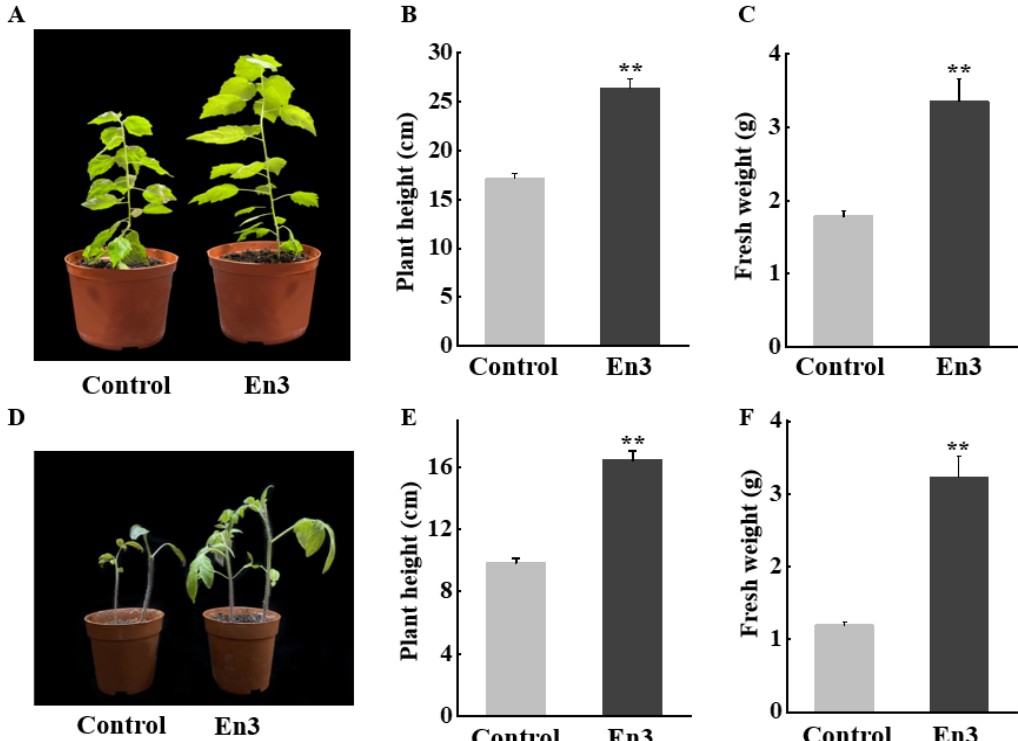

**Figure 1.** Plant growth-promoting activity of *Pseudomonas* sp. En3. (**A**) Morphology, (**B**) plant height, and (**C**) fresh weight of poplar 'Shanxin' were recorded after irrigation with an En3 suspension or water for one month. Data shown are mean ± SD (*n* = 12; ** *p* < 0.01 compared with control, Student's *t*-test). (**D**) Morphology, (**E**) plant height, and (**F**) fresh weight of tomato were recorded after irrigation with En3 suspension or water for two weeks. Values are means ± SD (*n* = 20; ** *p* < 0.01 compared with control, Student's *t*-test).

### 3.2. Analysis of PGP Features of Pseudomonas sp. En3

Phosphorus solubilization, nitrogen fixation, and IAA production are common features of PGP bacteria. The NBRIP medium, which incorporates insoluble $Ca_3(PO_4)_2$ as the sole phosphorus source, was used to evaluate the phosphorus solubilization capacity of strain En3. Following incubation at 28 °C for 5 days, colonies of *Pseudomonas* sp. En3 displayed a clear transparent halo around them, indicative of phosphorus solubilization (Figure 2A). In addition, En3 was able to grow well on nitrogen-free Ashby medium, suggesting its ability to fix atmospheric nitrogen (Figure 2B).

The Salkowski reagent was employed for the preliminary analysis of the capacity of strain En3 to synthesize indole derivatives. *P. syringae* pv. *tomato* DC3000, a model bacterial pathogen known for its ability to produce IAA, was selected as the positive control. The strains En3 and DC3000 were cultured to an equivalent concentration with an L-tryptophan supplement. Remarkably, both supernatants exhibited a discernible red color upon treatment with Salkowski's reagent; however, the supernatant of En3 displayed a more intense shade of red. Conversely, no obvious red coloration was observed in the supernatants of both En3 and DC3000 without L-tryptophan supplementation (Figure 2C). These findings suggest that En3 can produce indole derivatives in a tryptophan-dependent manner and to a greater extent than DC3000. To further resolve the indole derivatives produced by En3 and DC3000, we employed HPLC-MS/MS analysis to examine the supernatant of cultures with L-tryptophan supplement (Figures S1 and S2). Tables 2 and 3

demonstrate the presence of IAA in the culture supernatant of En3 and DC3000, with significantly higher levels observed in En3 (three biological replicates, $p < 0.01$), reaching a fold increase of 2.85. Notably, IAA biosynthesis intermediates (IAM, IPyA, and IAN) were also found at significantly higher levels in En3 compared to DC3000 (9.05-fold, 7.5-fold, and 8.81-fold, respectively).

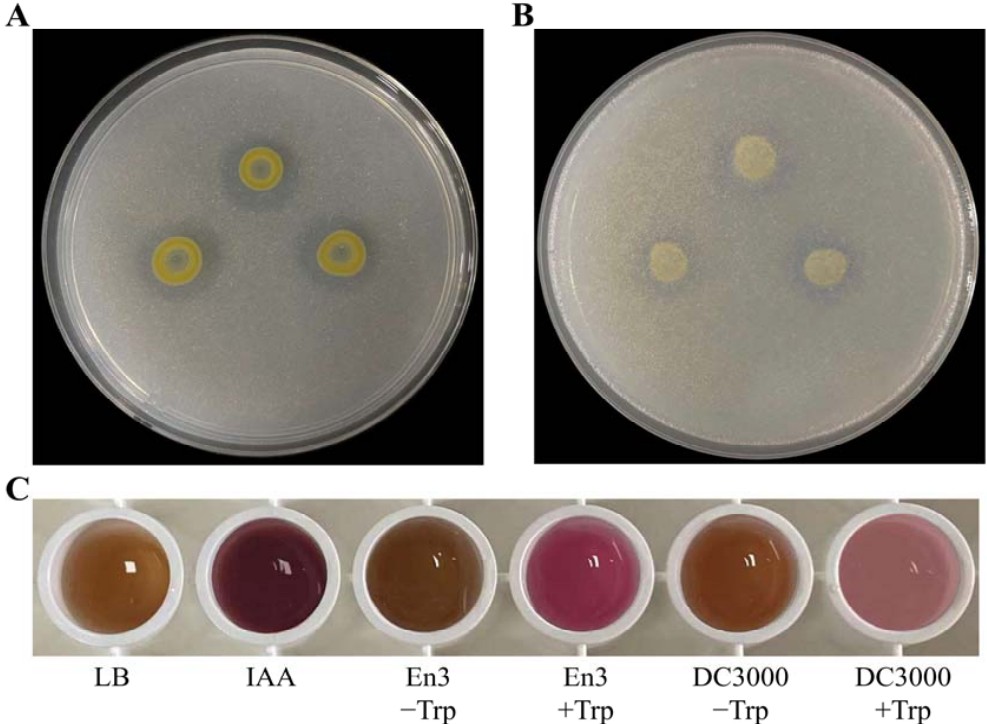

**Figure 2.** Characterization of *Pseudomonas* sp. En3 PGP features. (**A**) Phosphorous solution zones were observed when En3 was cultivated on NBRIP medium. (**B**) En3 grew on nitrogen-free Ashby medium. The photographs (**A**,**B**) were taken on the fifth day after inoculation. (**C**) Indole derivatives were detected using Salkowski reagent. LB, LB liquid medium; IAA, 50 mg/L IAA; En3 − Trp, culture supernatant of En3 without L-tryptophan; En3 + Trp, culture supernatant of En3 supplemented with 0.5 mg/mL L-tryptophan; DC3000 − Trp, culture supernatant of DC3000 without L-tryptophan; DC3000 + Trp, culture supernatant of DC3000 supplemented with 0.5 mg/mL L-tryptophan.

**Table 2.** HPLC-MS/MS analysis of indole derivatives produced by En3 in LB medium supplemented with L-tryptophan.

| Indole Derivatives | IAA | IAM | IPyA | IAN |
|---|---|---|---|---|
| Mass-to-charge ratio (M/Z) | 176.2 ≥ 129.8 | 175.0 ≥ 130.0 | 202.0 ≥ 114.9 | 157.1 ≥ 130.0 |
| Acqusion Standard | 5.49 | 4.74 | 6.10 | 6.35 |
| time(min) Sample | 5.50 | 4.74 | 6.10 | 6.35 |
| Concentration (ng/mL) [a] | 7.947 ± 0.342 | 1137.607 ± 21.849 | 1487.677 ± 37.020 | 14.100 ± 0.420 |

[a] Values are average ± SD ($n = 3$).

**Table 3.** HPLC-MS/MS analysis of indole derivatives produced by DC3000 in LB medium supplemented with L-tryptophan.

| Indole Derivatives | IAA | IAM | IPyA | IAN |
|---|---|---|---|---|
| Mass-to-charge ratio (M/Z) | 176.2 ≥ 129.8 | 175.0 ≥ 130.0 | 202.0 ≥ 114.9 | 157.1 ≥ 130.0 |
| Acqusion Standard | 5.49 | 4.74 | 6.10 | 6.35 |
| time(min) Sample | 5.51 | 4.74 | 6.10 | 6.35 |
| Concentration (ng/mL) [a] | 2.787 ± 0.214 | 125.743 ± 6.579 | 198.283 ± 12.689 | 1.600 ± 0.249 |

[a] Values are average ± SD ($n = 3$).

### 3.3. Analyzing Colonization Activity of Pseudomonas sp. En3

The seedlings of poplar 'Shanxin' and tomato showed no disease symptoms when continuously exposed to the En3 suspension. Hence, we aimed to further investigate whether direct infiltration of En3 into leaf tissues could induce disease symptoms and whether En3 could reproduce in plants. The infiltration of an En3 suspension of $10^5$ cfu/mL was performed via vacuum infiltration into micropropagated poplar seedlings, as well as via direct injection into leaves of tobacco, oilseed rape, and tomato. Plants were observed for 5 days. The infiltrated areas on the leaves of different plants retained their green coloration, while no significant changes were observed in the overall morphology of the infiltrated poplar plants (Figure 3A–D). Subsequently, a spontaneous mutant (En3$^{Rif}$) of En3 resistant to rifampicin was used for the analysis of bacterial populations grown in tobacco and poplar leaves. In poplar 'Shanxin', the En3$^{Rif}$ counts of 2 dpi (3.49 log cfu/mg) and 4 dpi (4.25 log cfu/mg) samples increased by 11- and 68-fold, respectively, compared to the En3$^{Rif}$ count of the 0 dpi (2.41 log cfu/mg) sample (Figure 3E). Similarly, the En3$^{Rif}$ counts in tobacco increased by 5-fold at 2 dpi (2.89 cfu/cm$^2$) and 12-fold at 4 dpi (3.23 cfu/cm$^2$) compared to 0 dpi (2.10 log cfu/cm$^2$) (Figure 3F). These findings suggest that the presence of En3 within the leaf tissue of multiple plant species does not induce disease symptoms and exhibits a capability for proliferation. The model pathogen *P. syringae* pv. *tomato* DC3000 is known to induce a hypersensitive response (HR) in *N. benthamiana*. Thus, we separately infiltrated *N. benthamiana* leaves with DC3000 and En3 at concentrations of $10^7$ cfu/mL and $10^5$ cfu/mL, respectively. It was observed that DC3000 triggered an HR, while En3 did not result in any symptoms (Figure S3), suggesting that the inoculation conditions used in this study effectively supported bacterial functionality.

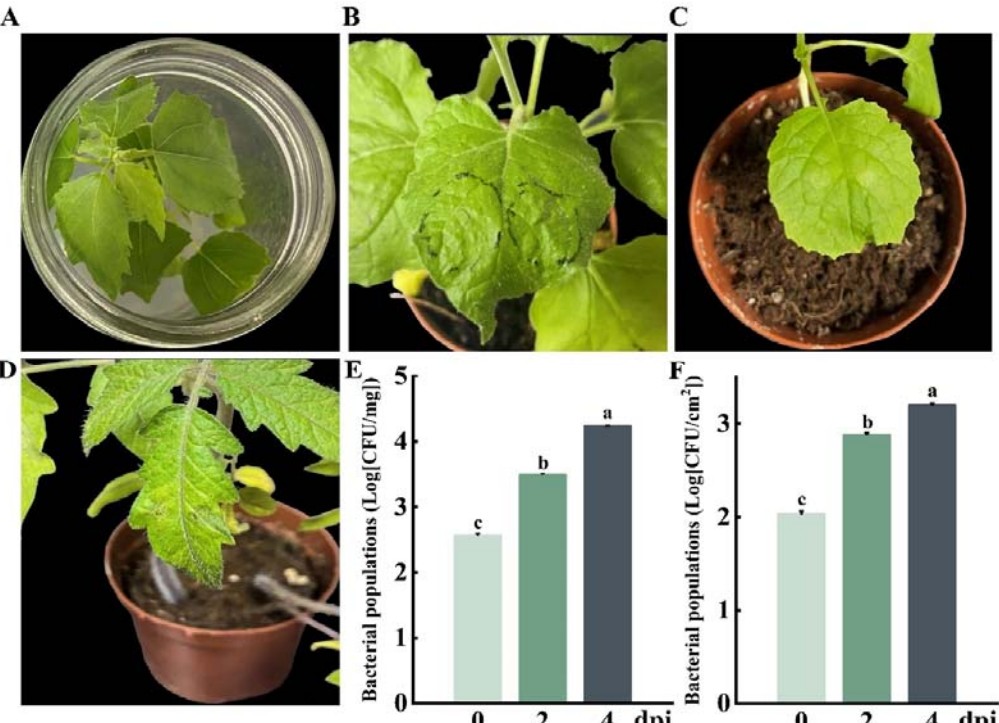

**Figure 3.** Colonization activity of *Pseudomonas* sp. En3. (**A–D**) Phenotypes of poplar 'Shanxin' (**A**), tobacco (**B**), oilseed rape (**C**), and tomato (**D**) leaves infiltrated with $10^5$ cfu/mL En3. Photographs were taken on 5 dpi. (**E,F**) Qualification of the En3$^{Rif}$ population in poplar 'Shanxin' (**E**) and tobacco (**F**). Different letters on the bars indicate significant differences ($p < 0.05$) (one-way ANOVA with three independent experiments).

### 3.4. General Features of the Pseudomonas sp. En3 Genome

Based on the trimmed Illumina (4.1 Gb) and ONT data (1.1 Gb), one circular chromosome of strain En3 of 5.35 Mb with 60.45% average G + C content was assembled. The sequencing depths were 207.14× (ONT) and 720.01× (Illumina). No plasmid was found. A total of 4807 protein-encoding sequences (CDSs) were predicted with an average length of 1003 bp, and this accounted for 90.1% of the total genome. In addition, the En3 genome had 67 tRNAs, 16 rRNA genes (5S, 16S, and 23S), 9 genomic islands, and 3 CRISPRs. The numbers of CDSs annotated in NR and KEGG databases were 4729 and 1746, respectively. A total of 501 predicted proteins had signal peptides and no transmembrane structures and were predicted to be secreted proteins (Table 4). The assembled and annotated sequences of En3 were deposited in GenBank (accession number CP124218).

**Table 4.** General features of the En3 genome.

| Feature | Value |
| --- | --- |
| Genome size (bp) | 5,350,973 |
| Contig | 1 |
| GC content (%) | 60.45 |
| tRNA | 67 |
| rRNA (5S, 16S, 23S) | 16 (6,5,5) |
| Protein-coding genes (CDS) | 4807 |
| Average CDS length (nt) | 1003 |
| CDS assigned to NR | 4729 |
| CDS assigned to KEGG | 1746 |
| Secreted protein | 501 |

### 3.5. Phylogenetic Analysis

A BLASTN search against the NCBI rRNA database using the 16S rDNA sequence (1347 bp) of strain En3 revealed that the top 10 hits were strains from the *P. fluorescens* group (6) or the *P. syringae* group (4), with approximate nucleotide differences (15–17 bp, identity > 99%). Phylogenetic analysis based on the 16S rRNA gene of representative type strains from various *Pseudomonas* groups indicated that strain En3 did not closely cluster with any known *Pseudomonas* species or group (Figure S4). To determine the taxonomic position of En3 accurately, an MLSA based on concatenated 16S rRNA gene and housekeeping genes *gyrB*, *rpoB*, and *rpoD* was performed. The strain En3 clustered with *P. rhizosphaerae* and *P. coleopterorum* with a high bootstrap support (81%) in the MLSA. Multiple taxonomic studies of the genus *Pseudomonas* have demonstrated that *P. rhizosphaerae* and *P. coleopterorum* constitute a distinct branch, separate from any defined group within the genus [4,29].

The average nucleotide identity (ANI) is widely used to determinate similarities between the genomes and an ANI value of 95% is considered as the cutoff for the delineation of bacterial species [33]. ANI analysis between the genome of *Pseudomonas* sp. En3 and that of *Pseudomonas* representative type strains showed that strain En3 exhibits a high ANI value with *P. rhizosphaerae* and *P. coleopterorum* (78.53% and 78.60%), while it has a lower value with bacteria from other clades (<78.28%). This further supports the inferred phylogenetic relationship based on multi-locus sequences. The ANI value between the genomes of *P. rhizosphaerae* and *P. coleopterorum* was 90.82%, much higher than that between En3 and *P. rhizosphaerae* and *P. coleopterorum*, indicating that strain En3 may represent a novel orphan species. In addition, an AutoMLST analysis based on 82 concatenated sequences of housekeeping genes from closely related genomes also supports the closest phylogenetic relationship between strain En3 and *P. coleopterorum* (Figure 4). A culture of strain En3 was deposited in the China Center for Type Culture Collection (AB 2023153).

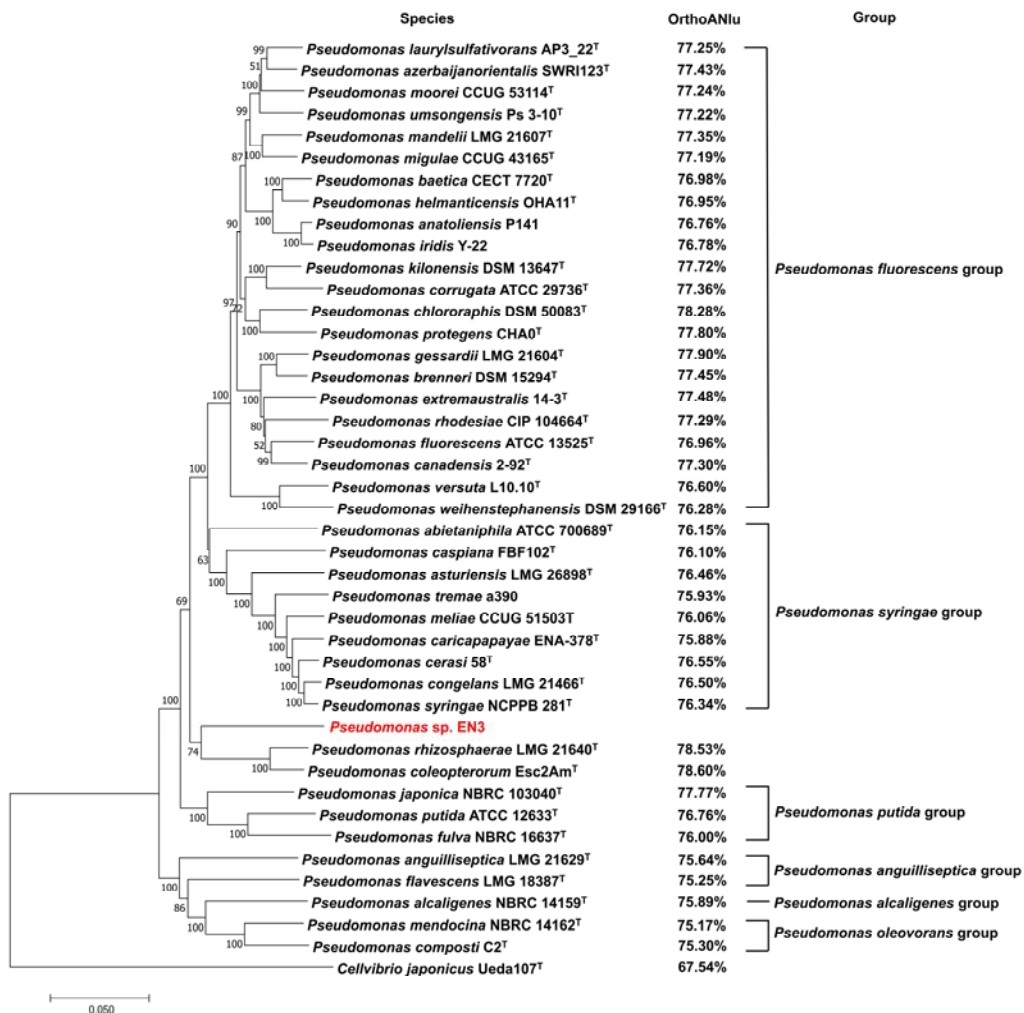

**Figure 4.** Neighbor-joining phylogenetic tree reconstruction based on the concatenated sequences of the four housekeeping universal genes (16S rRNA, *gyrB*, *rpoB*, and *rpoD*). Species, representative type strains from six *Pseudomonas* groups. OrthoANIu value, average nucleotide identity (ANI) based on a comparison of the En3 genome to that of Pseudomonas strains in the tree. Group, groups of *Pseudomonas*. Numbers at each node indicate the bootstrap percentage (*n* = 1000). The superscript letter "T" indicates the type strain.

### 3.6. Plant Growth-Promoting and Colonization Features of the Pseudomonas sp. En3 Genome

The gene annotation of En3 revealed that this bacterium possesses many genes and operons associated with plant growth-promoting activities. The En3 strains possess *iscU*, *iscS*, and *NRT* genes associated with nitrogen fixation; *gltB*, *gltD*, *nirB*, and *nirD* genes involved in ammonia assimilation; and *norR* and *hmp* genes related to nitrite stress for efficient nitrogen metabolism. For phosphorus metabolism, En3 has the PhoB/PhoR two-component system that senses the phosphorus concentration and the *phoU* and *pstABCS* operons involved in phosphate transport. We also observed the presence of the *ssuABCDE* operon, which is responsible for sulfonate transport, as well as the *cysAWUP* operon involved in sulfate transport, and *cysI* and *cysJ* genes associated with sulfite metabolism in the En3 genome. The En3 strain harbors genes responsible for the synthesis and transport of siderophores, including *entD*, *bfd*, *bfr*, *tonB*, *exbB*, *exbD*, and *fhuE*. As expected, the En3 strain harbors *trpABCDEG* operons and *iaaM* and *amiE* genes implicated in IAA biosynthesis. In addition, En3 possesses two ACC deaminases and one salicylate hydroxylase, which may be involved in the regulation of the plant hormones ethylene and salicylic acid (Table 5).

**Table 5.** Genes associated with PGP in the *Pseudomonas* sp. En3 genome.

| PGP Activity | Gene | Gene Annotation | Locus |
|---|---|---|---|
| Nitrogen Metabolism | *iscU* | nitrogen fixation protein NifU and related proteins | ctg_01045 |
| | *iscS* | cysteine desulfurase | ctg_01044 |
| | *NRT* | MFS transporter, nitrate/nitrite transporter | ctg_02133 |
| | *gltB* | glutamate synthase (NADPH) large chain | ctg_00450 |
| | *gltD* | glutamate synthase (NADPH) small chain | ctg_00451 |
| | *nirB* | nitrite reductase (NADH) large subunit | ctg_01700 |
| | *nirD* | nitrite reductase (NADH) small subunit | ctg_01701 |
| | *norR* | anaerobic nitric oxide reductase transcription regulator | ctg_01006 |
| | *hmp* | nitric oxide dioxygenase | ctg_01007 |
| Phosphate Metabolism | *phoB* | OmpR family, phosphate regulon response regulator | ctg_00055 |
| | *phoR* | OmpR family, phosphate regulon sensor kinase | ctg_00056 |
| | *phoU* | phosphate transport system protein | ctg_00061 |
| | *pstB* | phosphate transport system ATP-binding protein | ctg_00062 |
| | *pstA* | phosphate transport system permease protein | ctg_00063 |
| | *pstC* | phosphate transport system permease protein | ctg_00064 |
| | *pstS* | phosphate transport system substrate-binding protein | ctg_00065 |
| Sulfur metabolism | *ssuB* | sulfonate transport system ATP-binding protein | ctg_00353 |
| | *ssuC* | sulfonate transport system permease protein | ctg_00352 |
| | *ssuD* | alkanesulfonate monooxygenase | ctg_00351 |
| | *ssuA* | sulfonate transport system substrate-binding protein | ctg_00350 |
| | *ssuE* | FMN reductase | ctg_00349 |
| | *cysA* | sulfate transport system ATP-binding protein | ctg_00327 |
| | *cysW* | sulfate transport system permease protein | ctg_00326 |
| | *cysU* | sulfate transport system permease protein | ctg_00325 |
| | *cysP* | sulfate transport system substrate-binding protein | ctg_00324 |
| | *cysI* | sulfite reductase hemoprotein beta-component | ctg_02867 |
| | *cysJ* | sulfite reductase flavoprotein alpha-component | ctg_01699 |
| Siderophore | *tonB* | TonB-dependent transporters | ctg_00044, ctg_00494 |
| | *exbB* | biopolymer transport protein | ctg_03795 |
| | *exbD* | biopolymer transport protein | ctg_03794 |
| | *fhuE* | outer membrane receptor for ferric coprogen | ctg_03457, ctg_02420, ctg_02095, ctg_00748 |
| | *entD* | enterobactin synthetase component D | ctg_03886 |
| | *bfd* | bacterioferritin-associated ferredoxin | ctg_01441 |
| | *bfr* | bacterioferritin | ctg_01439, ctg_04538, ctg_00624 |
| Plant Hormones | *trpA* | tryptophan synthase alpha chain | ctg_00189 |
| | *trpB* | tryptophan synthase beta chain | ctg_00190 |
| | *trpE* | anthranilate synthase component I | ctg_04614 |
| | *trpG* | anthranilate synthase component II | ctg_04613 |
| | *trpD* | anthranilate phosphoribosyltransferase | ctg_04612 |
| | *trpC* | indole-3-glycerol phosphate synthase | ctg_04611 |
| | *iaaM* | tryptophan 2-monooxygenase | ctg_04661 |
| | *amiE* | amidase | ctg_02687, ctg_02400, ctg_03665 |
| | *NahG* | salicylate hydroxylase | ctg_02543 |
| | *dcyD* | 1-aminocyclopropane-1-carboxylate deaminase | ctg_03709, ctg_00163 |

As En3 can colonize multiple plant leaves, we conducted a genomic analysis of its features involved in host–plant interaction. The genome of En3 contains chemotaxis system genes that recognize environmental cues, including two *cheABDVWY* operons, one

*wspABCDEFR* operon, and 32 mcp protein genes. We identified a complete set of genes (*hcp*, *TssMLKJHGFECB*, *vgrG*, and *PAAR*) encoding the type VI secretion system in the En3 genome. The En3 strain also harbors gene clusters *hrpLJPVTGFDBZ* and *hrcVNRSTUJ*, which are responsible for the type III secretion system; however, it lacks *hrpA* and *hrpP* encoding the needle unit and needle-length regulator, respectively. Using the Effectidor prediction tool, seven type III secretion system effectors were predicted; three with annotations—avrE, CigR, and hopJ (Table S1).

## 4. Discussion

*Pseudomonas* represents one of the most diverse and ubiquitous bacterial genera among Gram-negative bacteria [34]. It encompasses a vast array of species, with 316 validly named species deposited in the List of Prokaryotic Names with Standing in Nomenclature (LPSN, https://lpsn.dsmz.de/genus/pseudomonas, accessed on 23 August 2023). Additionally, the number of *Pseudomonas* species in the LPSN database has consistently grown, as evidenced by the inclusion of 76 newly reported species in 2022. The 16S rRNA gene sequence is commonly used for the taxonomic classification of prokaryotes, although its discriminatory capacity often proves limited at the species level [35]. Multi-locus sequence analysis (MLSA), based on multiple housekeeping genes that are universal within the genus and have high sequence divergence, is, therefore, employed to elucidate the phylogenetic relationships among species within a given genus [36]. The commonly employed housekeeping genes for MLSA of the *Pseudomonas* genus include 16S rRNA, and the *rpoB*, *rpoD*, and *gryB* genes. Previous investigations have substantiated that the *Pseudomonas* phylogenetic groups defined via MLSA using these four housekeeping genes are concordant with outcomes derived from analyses involving a larger set of 100,120 housekeeping genes and whole-genome sequences [4].

In this study, we used the concatenated sequences of the four housekeeping genes from *Pseudomonas* sp. En3 and its closely related type strains to construct a phylogenetic tree. The type strains belonging to the *P. fluorescens*, *P. syringae*, *P. putida*, *P. anguilliseptica*, *P. oleovorans*, and *P. alcaligenes* groups were each clustered into distinct clades with robust bootstrap support. However, *Pseudomonas* sp. En3 was assigned within a distinct branch that includes *P. rhizosphaerae* and *P. coleopterorum*, which are not affiliated with any established *Pseudomonas* group. Furthermore, the results of OrthoANIu and AutoMLST based on a genome-wide comparison also supported that En3 had the highest similarity with *P. rhizosphaerae* and *P. coleopterorum*. The OrthoANIu value of En3 was found to be less than 78.60% when compared with all closely related *Pseudomonas* species, which is far below the cutoff of 95% ANI value used for species delineation [37]. Therefore, strain En3 can be classified as a novel *Pseudomonas* strain within the distinct branches of *P. rhizosphaerae* and *P. coleopterorum*.

The recent several taxonomic studies on *Pseudomonas* species provide support for the assignment of *P. rhizosphaerae* and *P. coleopterorum* as a distinct branch closely related to the groups of *P. putida*, *P. fluorescens*, *P. lutea*, and *P. syringae* [4,29,38]. The type strain LMG 21640[T] of *P. rhizosphaerae*, isolated from the rhizospheric soil of grass growing in Spain, has been confirmed to possess phosphate solubilizing activity [39]. The *P. coleopterorum* Esc2Am[T], which was isolated from the bark beetle (*Hylesinus fraxini*), has been found to possess cellulase-producing activity [40]. In this study, we identified a new member of this distinct branch from the leaf endosphere of poplar and subsequently demonstrated its growth-promoting activity for young seedlings of both poplar and tomato.

Phosphorus is an indispensable nutrient for the growth and productivity of plants, playing a crucial role in various vital plant functions such as photosynthesis, respiration, and energy transfer [41]. Most phosphorus, however, exists in soil as inorganic phosphate with a very low soluble concentration, rendering it unavailable to plants. The species of the *Pseudomonas* genus have been demonstrated to possess the ability to solubilize insoluble phosphates. Among them, *P. frederiksbergensis* JW-SD2 efficiently solubilizes phosphates and significantly promotes the growth of poplar trees [42]. In this study, we

demonstrated that *Pseudomonas* sp. En3 could efficiently dissolve tricalcium phosphate in the NBRIP medium, forming a clear dissolution zone. The analysis of the genome annotation further revealed that the En3 genome harbors a PhoB/PhoR two-component system, which functions as a phosphorus concentration sensor, and an integral pstABCS transporter responsible for facilitating phosphorus uptake. The speculation can be extended to suggest that En3 possesses the ability to assimilate diverse sources of phosphorus found in various environments.

Nitrogen is an essential element for plants, as it constitutes the fundamental components of nucleic acids and proteins. Despite its abundance in the atmosphere as nitrogen gas ($N_2$), it cannot be directly assimilated by plants [43]. Previous studies have confirmed that certain strains belonging to the *Pseudomonas* genus, such as *P. koreensis* CY4, *P. entomophila* CN11, *P. protegens* Pf-5 X940, and *P. stutzeri* A1501, possess nitrogen-fixing capabilities, which can enhance plant nutrient uptake and growth [44–46]. The growth of *Pseudomonas* sp. En3 on nitrogen-free Ashby medium suggests its capacity for atmospheric nitrogen fixation. Furthermore, genome annotation analysis revealed that En3 encodes iscU and iscS, which are thought to be involved in the assembly of [Fe-S] clusters of nitrogenase and are essential for nitrogen fixation [47]. The En3 genome also encodes multiple proteins involved in nitrogen metabolism, such as nirBD, which facilitates the conversion of nitrite to ammonia, and gltBD and hmp, which play a role in responding to nitrogen starvation [48]. The results obtained from genomic annotation and growth experiments provide support for the ability of the poplar endophytic bacterium En3 to solubilize phosphorus and fix nitrogen, indicating its potential as a biofertilizer for enhancing nutrient assimilation in forest and crop systems.

In this study, we discovered that strain En3 can enhance plant growth and colonize leaf tissues of various plants without inducing symptoms. The IAA produced by En3 may contribute to these two abilities. The major natural auxin of plants, IAA, is a crucial regulator of cell division, expansion, and differentiation, exerting significant influence on plant growth and development. Several PGP bacteria promote plant growth by producing IAA, for example, impairing the IAA synthesis of *P. putida* GR12-2 can diminish its capacity to enhance plant root growth [49]. On the other hand, IAA plays a pivotal role in mediating plant–microbe interactions. IAA can attenuate the salicylic acid-dependent immune response of plants and facilitate the colonization of certain biotrophic/hemi-biotrophic pathogens [50]. For instance, the disruption of the indole-3-acetaldehyde dehydrogenase genes resulted in impaired synthesis of IAA in *P. syringae* DC3000, leading to the reduced growth of DC3000 in *Arabidopsis thaliana* and the upregulation of the defense gene *PR1* [51].

The HPLC-MS/MS analysis in this study confirmed that strain En3 exhibited a significantly higher IAA production (2.85-fold) compared to DC3000 cultured under identical conditions, indicating that the concentration of IAA produced by En3 was sufficient to exert an inhibitory effect on the plant immune response. We also found that the levels of IAA biosynthesis intermediates (IAM, IPyA, and IAN) were significantly higher (>7.5-fold) in En3 compared to DC3000. In fact, the exogenous application of IAA intermediates can also elicit morphological alterations in plants; however, it remains unclear whether these alterations are attributable to the intermediates themselves or to IAA converted from these intermediates. For example, IAM treatment can induce *Arabidopsis* phenotypes resembling those of auxin overproduction mutants [52]. Therefore, the high levels of IAA and IAA intermediates in strain En3 may contribute to its plant growth-promoting activity.

Additionally, En3 harbors the genes encoding salicylate hydroxylase (NahG) and 1-aminocyclopropane-1-carboxylate (ACC) deaminases, which directly degrade salicylic acid and hydrolyze ethylene's precursor ACC, respectively. These genes may contribute to the asymptomatic colonization of En3 by regulating plant defense hormones. The En3 harbors components of the type III secretion system in its cytoplasm, inner membrane, periplasm, and outer membrane; however, it lacks the essential needle structure. This suggests that En3 may not possess the ability to inject effectors into plant cells but could potentially secrete effectors into the apoplast of plants to facilitate colonization. The type VI secretion

system of bacteria can deliver effectors into prokaryotic or eukaryotic cells, potentially facilitating manipulation of the host plant or competition with other microorganisms [53]. The investigation of whether the type VI secretion system of En3 possesses such functions warrants further exploration.

## 5. Conclusions

We isolated an endophytic bacterium, *Pseudomonas* sp. En3, from the leaf endosphere of poplar. Molecular phylogenetic and whole genome comparison analyses confirmed that strain En3 is a novel species in the genus *Pseudomonas*, distinct from any previously defined *Pseudomonas* groups. The strain En3 could significantly promote the growth of poplar and tomato and propagate in a variety of plant tissues without causing disease symptoms. The strain En3 exhibited significant growth-promoting activity on both poplar and tomato seedlings and demonstrated the ability to colonize the leaf tissue of multiple plants without inducing disease symptoms. We conducted experimental validation to confirm the capabilities of En3 in phosphorus solubilization, nitrogen fixation, and IAA production. The analysis of the genome annotation of En3 unveiled a series of crucial genes and pathways associated with promoting plant growth. These findings provide support for the potential use of the En3 strain as a biofertilizer for various crops and trees.

**Supplementary Materials:** The following supporting information can be downloaded at: https://www.mdpi.com/article/10.3390/f14112203/s1, Figure S1: Mass spectra of IAA and IAM produced by En3 and DC3000; Figure S2: Mass spectrum of IAA synthesis intermediates produced by En3 and DC3000; Figure S3: *N. benthamiana* leaves infiltrated with different concentrations of *P. syringae* pv. *tomato* DC3000 and *Pseudomonas* sp. En3; Figure S4: Neighbor-joining phylogenetic tree reconstruction based on 16S rRNA genes; Table S1: Genes associated with colonization activity in *Pseudomonas* sp. En3 genome.

**Author Contributions:** Conceptualization, B.D., L.W., H.X. and Q.C.; methodology, B.D.; software, B.D.; validation, B.D., L.W. and H.X.; formal analysis, B.D.; investigation, B.D.; resources, Q.C.; data curation, B.D.; writing—original draft preparation, B.D.; writing—review and editing, Q.C.; visualization, Q.C.; supervision, Q.C.; project administration, Q.C.; funding acquisition, Q.C. All authors have read and agreed to the published version of the manuscript.

**Funding:** This work was supported by the National Natural Science Foundation of China (Grant No. 31870658).

**Data Availability Statement:** The assembled sequences of En3 were deposited in GenBank (accession number CP124218) and the Genome Warehouse in the National Genomics Data Center (NGDC) (accession number GWHEQCH00000000). A culture of strain En3 was deposited in the China Center for Type Culture Collection (AB 2023153).

**Conflicts of Interest:** The authors declare no conflict of interest.

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
