# Peer review of "Characterization of Pseudomonas sp. En3, an Endophytic Bacterium from Poplar Leaf Endosphere with Plant Growth-Promoting Properties"

_forests, doi:10.3390/f14112203_

Round 1

Reviewer 1 Report

Comments and Suggestions for Authors

The manuscript entitled”Characterization of Pseudomonas sp. En3, an endophytic bacterium from poplar leaf endosphere with plant growth-promoting properties” demonstrated Pseudomonas sp. En3 had significant growth-promoting effects on both poplar and tomato seedlings. It would be a worthy value for Forests. Nevertheless, I have some major concerns:

1. About the 16srDNA sequence identification of strain EN3. In line 179, Pseudomonas baltica strain MBT-2 (98.62% identity) . However, I used the 16s sequence of strain EN3 (OR511655) to BLAST against the NCBI and ezBiocloud (https://www.ezbiocloud.net/). The results showed that strain EN3 were closely related to Pseudomonas azerbaijanorientalis SWRI123T with 99.10%, followed by Pseudomonas iridis P42T (98.94%), Pseudomonas anatoliensis P9T (98.94%) and Pseudomonas moorei RW10T (98.81%). All these results were higher than the novel species boundary 98.8%, but these strains above did not include in this paper. So, authors need to re-describe the 16s alignment results in this paper and reconstruct the phylogenetic trees with outgroup.

2. I did not find the genome sequence of strain EN3 published on NCBI (CP124218). Therefore, the authors need to to re-select strains based on the 16S alignment results to calculate the ANI value. If the ANI value is lower than the species boundary (95.0-96.0%) , the author can consider the strain as a potential new species.

3. The names of type strain in these paper were not standardized.If a strain was type strain, please superscript T.

4. In the part of 2.5, please provide references.

5. In the part of 3.2, authors use P. syringae pv. tomato DC3000 as the control. However, in part 3.3, authors only use strain EN3 to show that there was no disease symptoms on plants leaves. Why not use DC3000 as a control?

6. In the genome of strain EN3, authors identified a complete set of genes of type III and VI secretion system, which mainly related to pathogenicity. Are there any genetic differences between strain EN3 (plant growth promoting) and DC3000 (pathogen)? I suggest the author compare and discuss in the article.

Minor comments:

1. Table 3, What does Title2 mean?

2. Line 198,192 and entire text, P < 0.01, P should be italic.

3. The writing of gene names in the entire text is not standardized, please use italics.

Comments on the Quality of English Language

Minor editing of English language required

Author Response

         Thanks for your comments concerning our manuscript entitled “Characterization of Pseudomonas sp. En3, an endophytic bacterium from poplar leaf endosphere with plant growth-promoting properties”. Those comments are all valuable and helpful for revising and improving our paper. According with your advice, we have made modification (red font) on the original manuscript. Please see below, in blue, for a point-by-point response to the reviewers' comments and concerns.

  1. About the 16srDNA sequence identification of strain EN3. In line 179, Pseudomonas baltica strain MBT-2 (98.62% identity) . However, I used the 16s sequence of strain EN3 (OR511655) to BLAST against the NCBI and ezBiocloud (https://www.ezbiocloud.net/). The results showed that strain EN3 were closely related to Pseudomonas azerbaijanorientalis SWRI123T with 99.10%, followed by Pseudomonas iridis P42T (98.94%), Pseudomonas anatoliensis P9T (98.94%) and Pseudomonas moorei RW10T (98.81%). All these results were higher than the novel species boundary 98.8%, but these strains above did not include in this paper. So, authors need to re-describe the 16s alignment results in this paper and reconstruct the phylogenetic trees with outgroup.

Response:

         Thank you for pointing out that we did not describe this adequately. We have made revisions to address this issue.

 (1) The ezBiocloud platform indeed provides more information, so we have made modifications to the result of 16S alignment in result 1.

 (2) We reconstructed the phylogenetic trees of 16S and MLSA using additional data from Pseudomonas azerbaijanorientalis SWRI123T, Pseudomonas moorei CCUG 53114T, Pseudomonas iridis Y-22, and Pseudomonas anatoliensis P141.

 (3) We added outgroup to phylogenetic analysis (Cellvibrio japonicus Ueda107T from Gammaproteobacteria).

Note:

         The 16S rDNA sequence (AM293566) of Pseudomonas moorei RW10T contains 18 degenerate bases; therefore, we opted to utilize the sequences from Pseudomonas moorei CCUG 53114T for conducting the phylogenetic analysis.

         Due to the unavailability of the genome sequence for Pseudomonas iridis P42T, we chose the sequences from Pseudomonas iridis Y-22 for conducting the phylogenetic analysis. The 16S rDNA sequence of P42T and Y-22 exhibit a complete match with 100% identity.

         Due to the unavailability of the genome sequence for Pseudomonas anatoliensis P9T, we chose the sequences from Pseudomonas anatoliensis P141 for conducting the phylogenetic analysis. The 16s rDNA sequence of P9T and P141 exhibit a complete match with 100% identity.

  1. I did not find the genome sequence of strain EN3 published on NCBI (CP124218). Therefore, the authors need to to re-select strains based on the 16S alignment results to calculate the ANI value. If the ANI value is lower than the species boundary (95.0-96.0%) , the author can consider the strain as a potential new species.

Response:

         We gratefully appreciate for your comment. In fact, the En3 assembly is scheduled to be released on April 10, 2024, or upon publication of the GenBank Accession Number. Recently, I reached out to NCBI requesting immediate release of CP124218. Additionally, we have submitted the assembly to the Genome Warehouse at the National Genomics Data Center (NGDC) with accession number GWHEQCH00000000. Currently, the assembly stored in NGDC is readily available.

          The web service of JSpecies (http://jspecies.ribohost.com/jspeciesws/) was not functioning until today, preventing us from calculating the ANI value between the five newly added bacteria and En3 using the exactly same method as before. Instead, we utilized ezBiocloud's OrthoANI tool to calculate the OrthoANIu value. As anticipated, there is a strong correlation between ezBiocloud's OrthoANIu and JSpecies’ ANIb values (table below). We therefore employed the ANI values generated by OrthoANI in our text. The ANI values detected are obviously lower than the 95% threshold, suggesting that En3 is likely a new species.

ANIb of JSpecies

OrthoANIu of ezBiocloud's

En3 vs. Cellvibrio japonicus Ueda107T

NA

67.54%

En3 vs. Pseudomonas anatoliensis P141

NA

76.76%

En3 vs. Pseudomonas azerbaijanorientalis SWRI123T

NA

77.43%

En3 vs. Pseudomonas iridis Y-22

NA

76.78%

En3 vs. Pseudomonas moorei CCUG 53114T

NA

77.24%

En3 vs. Pseudomonas abietaniphila ATCC 700689T

75.17%

76.15%

En3 vs. Pseudomonas alcaligenes NBRC 14159T

74.09%

75.89%

En3 vs. Pseudomonas anguilliseptica LMG 21629T

73.98%

75.64%

En3 vs. Pseudomonas asturiensis LMG 26898T

75.16%

76.46%

En3 vs. Pseudomonas baetica CECT 7720T

75.73%

76.98%

En3 vs. Pseudomonas brenneri DSM 15294T

75.98%

77.45%

En3 vs. Pseudomonas canadensis 2-92T

75.85%

77.30%

En3 vs. Pseudomonas caricapapayae ENA-378T

74.90%

75.88%

En3 vs. Pseudomonas caspiana FBF102T

75.14%

76.10%

En3 vs. Pseudomonas cerasi 58T

75.40%

76.55%

En3 vs. Pseudomonas chlororaphis DSM 50083T

76.76%

78.28%

En3 vs. Pseudomonas coleopterorum Esc2AmT

77.16%

78.60%

En3 vs. Pseudomonas composti C2T

73.56%

75.30%

En3 vs. Pseudomonas congelans LMG 21466T

75.35%

76.50%

En3 vs. Pseudomonas corrugata ATCC 29736T

75.68%

77.36%

En3 vs. Pseudomonas extremaustralis 14-3T

76.25%

77.48%

En3 vs. Pseudomonas flavescens LMG 18387T

73.71%

75.25%

En3 vs. Pseudomonas fluorescens ATCC 13525T

75.81%

76.96%

En3 vs. Pseudomonas fulva NBRC 16637T

74.56%

76.00%

En3 vs. Pseudomonas gessardii LMG 21604T

76.25%

77.90%

En3 vs. Pseudomonas helmanticensis OHA11T

75.59%

76.95%

En3 vs. Pseudomonas japonica NBRC 103040T

76.32%

77.77%

En3 vs. Pseudomonas kilonensis DSM 13647T

76.36%

77.72%

En3 vs. Pseudomonas laurylsulfativorans AP3_22T

75.95%

77.25%

En3 vs. Pseudomonas mandelii LMG 21607T

75.84%

77.35%

En3 vs. Pseudomonas meliae CCUG 51503T

74.94%

76.06%

En3 vs. Pseudomonas mendocina NBRC 14162T

73.48%

75.17%

En3 vs. Pseudomonas migulae CCUG 43165T   

75.86%

77.19%

En3 vs. Pseudomonas protegens CHA0T

76.46%

77.80%

En3 vs. Pseudomonas putida ATCC 12633T

75.51%

76.76%

En3 vs. Pseudomonas rhizosphaerae LMG 21640T

77.27%

78.53%

En3 vs. Pseudomonas rhodesiae CIP 104664T

75.89%

77.29%

En3 vs. Pseudomonas syringae NCPPB 281T

75.26%

76.34%

En3 vs. Pseudomonas tremae a390

74.67%

75.93%

En3 vs. Pseudomonas umsongensis Ps 3-10T

75.93%

77.22%

En3 vs. Pseudomonas versuta L10.10T

75.00%

76.60%

En3 vs. Pseudomonas weihenstephanensis DSM 29166T

74.89%

76.28%

  1. The names of type strain in these paper were not standardized. If a strain was type strain, please superscript T.

Response:

        Thanks for your suggestion. We have superscripted all the type strains with T in the revised manuscript.

  1. In the part of 2.5, please provide references.

Response:

        As suggested by the reviewer, we have added references to support this idea.

[25] Sun K.; Liu J.; Gao Y.Z.; Sheng Y.H.; Kang F.X.; Waigi M.G. Inoculating plants with the endophytic bacterium Pseudomonas sp. Ph6-gfp to reduce phenanthrene contamination. Environ. Sci. Pollut. Res 201522, 19529-19537.
  1. In the part of 3.2, authors use P. syringae pv. tomato DC3000 as the control. However, in part 3.3, authors only use strain EN3 to show that there was no disease symptoms on plants leaves. Why not use DC3000 as a control?

Response:

        Thank you for your comment. The model pathogen P. syringae pv. tomato DC3000 is known to cause disease in tomatoes and Arabidopsis, as well as induce a hypersensitive response (HR) in Nicotiana benthamiana. Thus, we separately inoculated N. benthamiana leaves with DC3000 and En3 at concentrations of 107 cfu/mL and 105 cfu/mL, respectively. We have added a supplementary figure showing the infiltration of DC3000 and En3 into N. benthamiana leaves. It was observed that DC3000 triggered an HR, while En3 did not result in any symptoms. These findings suggest that the inoculation conditions used in this study effectively supported bacterial functionality.

Figure S3. N. benthamiana leaves infiltrated with different concentrations of P. syringae pv. tomato DC3000 and Pseudomonas sp. En3. (A) The concentration of DC3000 was 107 cfu/mL. (B) The concentration of DC3000 was 105 cfu/mL. (C) The concentration of EN3 was 107 cfu/mL. (D) The concentration of EN3 was 105 cfu/mL. Photograph was taken on 2 dpi.

  1. In the genome of strain EN3, authors identified a complete set of ge of type III and VI secretion system, which mainly related to pathogenicity. Are there any genetic differences between strain EN3 (plant growth promoting) and DC3000 (pathogen)? I suggest the author compare and discuss in the article.

Response:

        Thank you for the suggestion to improve the paper. The strain En3 possesses an almost complete type III secretion system, including components in its cytoplasm, inner membrane, periplasm, and outer membrane. However, it lacks hrpA and hrpP genes encoding the needle unit and needle-length regulator, respectively. To provide further clarification on this matter, we have made revisions to the manuscript.

“The En3 strain also harbors gene clusters hrpLJPVTGFDBZ and hrcVNRSTUJ, which are responsible for the type III secretion system; however, it lacks hrpA and hrpP encoding the needle unit and needle-length regulator, respectively.”

        We also added a discussion at the end of the manuscript.

“The En3 harbors components of the type III secretion system in its cytoplasm, inner mem-brane, periplasm, and outer membrane; however, it lacks the essential needle structure. This suggests that En3 may not possess the ability to inject effectors into plant cells but could potentially secrete effectors into the apoplast of plants to facilitate colonization. The type VI secretion system of bacteria can deliver effectors into prokaryotic or eukaryotic cells, potentially facilitating manipulation of the host plant or competition with other mi-croorganisms. The investigation of whether the type VI secretion system of En3 possesses such functions warrants further exploration.”

Minor comments:

  1. Table 3, What does Title2 mean?

Response:

        We sincerely thank the reviewer for careful reading. Since we have added a Table 1, Table 3 becomes Table 4. We have corrected the “Title2” into “Value”.

  1. Line 198,192 and entire text, P < 0.01, P should be italic.

Response:

        We were really sorry for our careless mistakes. Thank you for your reminder. We have standardized the writing of the P in the paper, using italics.

  1. The writing of gene names in the entire text is not standardized, please use italics.

Response:

       Thanks for your careful checks. We are sorry for our carelessness. Based on your comments, we have italicised the gene names in the manuscript.

        We tried our best to improve the manuscript and made some changes marked in red in revised paper which will not influence the content and framework of the paper. We appreciate for Editors/Reviewers’ warm work earnestly, and hope the correction will meet with approval. Once again, thank you very much for your comments and suggestions.

Sincerely,

Qiang Cheng

Reviewer 2 Report

Comments and Suggestions for Authors

Add more bibliographic citations of species of the genus Pseudumonas with plant growth promoting activity

What is the concentration of bacteria of the four strains inoculated in the seeds?

I suggest placing the 16s gene identity results of the strains in a table.

In the methodology section where the primers are mentioned, add the bibliography from which they were consulted or mention if they were designed by the authors.

In the plant growth promotion bioassays, Solanum lycopersicum seeds were used.

Comments on the Quality of English Language

Minor editing of English language required

Author Response

        Thanks for your comments concerning our manuscript entitled “Characterization of Pseudomonas sp. En3, an endophytic bacterium from poplar leaf endosphere with plant growth-promoting properties”. Those comments are all valuable and helpful for revising and improving our paper. We have studied all comments carefully and have made conscientious correction. According with your advice, we have made modification (red font) on the original manuscript. Please see below, in blue, for a point-by-point response to the reviewers' comments and concerns.

      1.Add more bibliographic citations of species of the genus Pseudumonas with plant growth promoting activity.

Response:

        We sincerely appreciate the valuable comments. We have checked the literature carefully and added more references on plant growth-promoting Pseudomonas into the introduction and discussion part in the revised manuscript.

Introduction part :

“The endophytic bacterium P. brassicacearum Zy-2-1 from Sphaerophysa salsula, P. fluorescens Sasm05 from Sedum alfredii Hance, and P. azotoformans ASS1 from Alyssum serpyllifolium can protect plants from abiotic stress and expedite the phytoremediation process of metal-contaminated soils [13-15].”

[13] Kong Z.Y.; Deng Z.S.; Glick B.R.; Wei G.H.; Chou M.X. A nodule endophytic plant growth-promoting Pseudomonas and its effects on growth, nodulation and metal uptake in Medicago lupulina under copper stress. Ann. Microbiol 201767, 49-58.
[14] Chen B.; Luo S.; Wu Y.J.; Ye J.Y.; Wang Q.; Xu X.M.; Pan F.S.; Khan K.Y.; Feng Y.; Yang X. The effects of the endophytic bacterium Pseudomonas fluorescens Sasm05 and IAA on the plant growth and cadmium uptake of Sedum alfredii Hance. Front. Microbiol 20178, 2538.
[15] Ma Y.; Rajkumar M.; Moreno A.; Zhang C.; Freitas H. Serpentine endophytic bacterium Pseudomonas azotoformans ASS1 accelerates phytoremediation of soil metals under drought stress. Chemosphere 2017, 185, 75-85.

Discussion part :

“Species of the Pseudomonas genus have been demonstrated to possess the ability to solubilize insoluble phosphates. Among them, P. frederiksbergensis JW-SD2 efficiently solubilizes phosphates and significantly promotes the growth of poplar trees [42].”

[42]  Zeng Q.W.; Wu X.Q.; Wen X.Y. Identification and characterization of the rhizosphere phosphate-solubilizing bacterium Pseudomonas frederiksbergensis JW-SD2, and its plant growth-promoting effects on poplar seedlings. Ann. Microbiol 2016, 66, 1343-1354.

“Previous studies have confirmed that certain strains belonging to the Pseudomonas genus, such as P. koreensis CY4, P. entomophila CN11, P. protegens Pf-5 X940, and P. stutzeri A1501, possess nitrogen-fixing capabilities, which can enhance plant nutrient uptake and growth [44-46].”

[44]  Li H.B.; Singh R.K.; Singh P.; Song Q.Q.; Xing Y.X.; Yang L.T.; Li Y.R. Genetic diversity of nitrogen-fixing and plant growth promoting Pseudomonas species isolated from sugarcane rhizosphere. Front. Microbiol 20178, 1268.
[45]  Fox A.R.; Soto G.; Valverde C.; Russo D.; Lagares Jr A.; Zorreguieta Á.; Alleva K.; Pascuan C.; Frare R.; Ayub N.D. Major cereal crops benefit from biological nitrogen fixation when inoculated with the nitrogen‐fixing bacterium Pseudomonas protegens Pf‐5 X940.  Environ. Microbiol 201618, 3522-3534.
[46]  Ke X.B.; Feng S.; Wang J.; Lu W.; Zhang W.; Chen M.; Lin M. Effect of inoculation with nitrogen-fixing bacterium Pseudomonas stutzeri A1501 on maize plant growth and the microbiome indigenous to the rhizosphere. Syst. Appl. Microbiol 2019, 42, 248-260.
  1. What is the concentration of bacteria of the four strains inoculated in the seeds?

Response:

       In line 105, we describe that we are using four bacterial solutions OD600=0.5 concentrations to water seedlings to explore whether they can promote plant growth.

  1. I suggest placing the 16s gene identity results of the strains in a table.

Response:

       Thanks for your suggestion. We think this is an excellent suggestion. We tabulated the 16s rRNA identifications of the four endophytes isolated from poplar leaves and attached them to the revised manuscript.

Table 1Sequence analysis of 16S rRNA of endophytic bacteria isolated from Poplar leaves.

Endophyte

Top hit (accession No.)

Sequence length (bp)

% Identity

Genbank accession No.

En1

Sphingomonas sanguinis NBRC 13937T (NR_113637)

1409

99.86

OR511661

En2

Sphingobium yanoikuyae NBRC 15102T (NR_113730)

1409

99.72

OR511660

En3

Pseudomonas azerbaijanorientalis SWRI123T (CP077078)

1448

99.10

OR511655

En4

Microbacterium testaceum NBRC 12675T (NZ_BJMLO1000022)

1444

99.10

OR511666

  1. In the methodology section where the primers are mentioned, add the bibliography from which they were consulted or mention if they were designed by the authors.

Response:

        We sincerely appreciate the valuable comments. As suggested by the reviewer, we have added references to support this idea.

[21]  Prodhan M.Y.; Rahman M.B.; Rahman A.; Akbor M.A.; Ghosh S.; Nahar, M.N.E.; Simo.; Shamsuzzoha M.; Cho K.M.; Haque M.A.  Characterization of Growth-Promoting Activities of Consortia of Chlorpyrifos Mineralizing Endophytic Bacteria Naturally Harboring in Rice Plants—A Potential Bio-Stimulant to Develop a Safe and Sustainable Agriculture. Microorganisms 202311, 1821.
  1. In the plant growth promotion bioassays, Solanum lycopersicum seeds were used.

Response:

       Thank you for your comment. For clarification, we revised Tomato plants (Solanum lycopersicum) were sowed under identical conditions in the same growth chamber; one-week-old tomato seedlings were used for plant growth-promoting assays.

       We tried our best to improve the manuscript and made some changes marked in red in revised paper which will not influence the content and framework of the paper. We appreciate for Editors/Reviewers’ warm work earnestly, and hope the correction will meet with approval. Once again, thank you very much for your comments and suggestions.

Sincerely,

Qiang Cheng

Round 2

Reviewer 1 Report

Comments and Suggestions for Authors

Usually, ‘novel species' is more commonly used than 'new species'. So, I suggest using 'novel species' in the whole paper. 

Accept